# Characterization of Residual Stress in SOI Wafers by Using MEMS Cantilever Beams

**DOI:** 10.3390/mi14081510

**Published:** 2023-07-27

**Authors:** Haotian Yang, Min Liu, Yingmin Zhu, Weidong Wang, Xianming Qin, Lilong He, Kyle Jiang

**Affiliations:** 1School of Mechano-Electronic Engineering, Xidian University, Xi’an 710071, China; yanghaotian_sweet@163.com (H.Y.); mliu_12@stu.xidian.edu.cn (M.L.); ymzhu@xidian.edu.cn (Y.Z.); qinxianming@xidian.edu.cn (X.Q.); k.jiang@bham.ac.uk (K.J.); 2City U-Xidian Joint Laboratory of Micro/Nano-Manufacturing, Shenzhen 518057, China; 3Xi’an Chuanglian Electronic Component (Group) Co., Ltd., Xi’an 710065, China; lilonghe1980@163.com; 4School of Mechanical Engineering, University of Birmingham, Birmingham B15 2TT, UK

**Keywords:** SOI, residual stress, cantilever beam, MEMS, characterization

## Abstract

Silicon-on-insulator (SOI) wafers are crucial raw materials in the manufacturing process of microelectromechanical systems (MEMS). Residual stresses generated inside the wafers during the fabrication process can seriously affect the performance, reliability, and yield of MEMS devices. In this paper, a low-cost method based on mechanical modeling is proposed to characterize the residual stresses in SOI wafers in order to calculate the residual stress values based on the deformation of the beams. Based on this method, the residual strain of the MEMS beam, and thus the residual stress in the SOI wafer, were experimentally determined. The results were also compared with the residual stress results calculated from the deflection of the rotating beam to demonstrate the validity of the results obtained by this method. This method provides valuable theoretical reference and data support for the design and optimization of devices based on SOI-MEMS technology. It provides a lower-cost solution for the residual stress measurement technique, making it available for a wide range of applications.

## 1. Introduction

The residual stress in silicon-on-insulator (SOI) wafers has been a significant concern impacting the reliability of microsystem devices and the performance of MEMS devices, potentially leading to device failure [1]. Therefore, it is essential to study the residual stress in MEMS devices. For devices based on SOI-MEMS technology, the residual stress primarily arises from two sources: residual stress in the SOI wafer and residual stress formed during the processing of MEMS devices. While previous research has focused on residual stress in thin layers [2,3,4,5], or residual stress generated during processing [6,7,8,9], there is limited research on the residual stress in SOI chips themselves.

Residual stress testing methods can be broadly categorized as either destructive or non-destructive. Destructive methods release stress by mechanically separating or splitting the components on the surface. Recently, optical techniques [10,11,12,13], digital image processing [14,15,16], and incremental hole-drilling methods [17] have been extended to characterize residual stresses with high accuracy. However, these techniques are not suitable for measuring residual stresses in micro and nano devices or SOI-MEMS devices due to their destructive nature, which can cause material damage and yielding.

Non-destructive measurement methods for residual stress have made significant progress in recent years. These include radiation methods [18,19], nanoindentation strain methods [20], and residual strain measurement methods [21,22], among others. For example, E. Dobročka et al. [18] developed a method to measure residual stress in strongly woven ZnO layers using X-ray diffraction stress analysis, achieving high accuracy. However, radiation methods suffer from shallow measurement depth and high costs. On the other hand, nanoindentation is a popular technique due to its low damage and ease of measurement. However, different theoretical calculation models used in nanoindentation yield different results in mechanical energy properties such as contact area, material hardness, and stress. Commonly used nanoindentation theory calculation models include the Suresh theory model [23], Lee model [24], Swadener theory model [25], and Xu model [26].

To overcome these challenges, this study introduces a non-contact method to characterize the residual stress in SOI-MEMS devices. The residual strain measurement method calculates the residual stress value by measuring the residual strain and applying theoretical calculations based on mechanical equations. As a non-destructive measurement method, it has received much attention in recent years due to its scalability. The proposed method in this paper utilizes theoretical calculations and bending moment measurements to accurately assess the residual stress. By examining the underlying mechanisms of residual stress generation in SOI wafers, mechanical deflection models for MEMS cantilever beams and fixed beams are developed. Experimental testing is conducted to validate the effectiveness and feasibility of this method for characterizing residual stress in SOI wafers.

## 2. Mechanism Analysis of Residual Stress in SOI Wafer

Silicon direct bonding (SDB) technology is widely used in the preparation of SOI wafers. The process involved in creating bonded SOI wafers is illustrated in Figure 1 [27,28]. It typically begins with a wet chemical and ultrasonic cleaning of the silicon wafers, which include the device wafer and the handle wafer. Following this, the device layer undergoes thermal oxidation to create a surface oxide layer, which ultimately becomes the buried oxide in the SOI structure. The two wafers are then pre-bonded in an ultra-clean environment. The next step involves annealing the bonded wafer pairs at temperatures in the range of 1100–1200 °C. This process promotes the formation of strong Si-O-Si or Si-Si bonds, ensuring a permanent bond between the wafers. Finally, the device layer is polished to achieve the desired thickness.

The SDB technology offers several advantages in the preparation of SOI wafers. It allows for the creation of a thin and defect-free buried oxide layer, which is crucial for the performance of SOI devices. The bonding process enables the integration of different materials and the formation of high-quality interfaces. Additionally, the ability to control the thickness of the device layer through polishing allows for customization according to specific device requirements.

The process of thermal oxidation, which involves growing a SiO_2_ layer on the surface of a polished silicon wafer, induces the generation of residual stresses within the layer. These residual stresses can be categorized into thermal stress and intrinsic stress [22,23]. Thermal stress, also known as thermal mismatch stress, arises due to the disparity in the thermal expansion coefficients between the layer and the substrate material. On the other hand, intrinsic stress is more complex in nature, primarily attributed to the mismatch in lattice structures between the layer and the substrate material [4], as depicted in Figure 2 [29]. The lattice mismatch leads to the formation of edge dislocations, and these dislocations create an associated elastic stress field around them.

During the thermal growth of the SiO_2_ layer on the polished silicon wafer surface, a significant amount of oxygen atoms is introduced into the silicon surface, leading to volume expansion. However, since the grown SiO_2_ layer is bonded to the substrate, its perimeter cannot freely expand. Considering that the thickness of the substrate is much larger than that of the SiO_2_ layer, it can be assumed that the radial expansion of the SiO_2_ layer is nearly zero. As a result, the SiO_2_ layer can only expand in the direction perpendicular to the substrate surface, resulting in the presence of intrinsic compressive stress distributed in a gradient along the thickness direction of the SiO_2_ layer.

During the subsequent cooling process, the SiO_2_ layer experiences a uniformly distributed thermal mismatch stress. This is because the thermal expansion coefficient of SiO_2_ is lower than that of silicon. After the growth of the SiO_2_ layer, the residual stress within the layer is a combination of the thermal mismatch stress and the intrinsic stress. At this point, the SiO_2_ layer and the silicon substrate reach equilibrium, and tensile stress is present in the substrate. The distribution of residual stress in the layer and substrate is illustrated in Figure 3.

In the prepared SOI wafer, the distribution of residual stress within the wafer is depicted in Figure 4. The SiO_2_ layer experiences compressive stress from the underlying silicon and the device layer, while the underlying silicon and the device layer undergo tensile stress from the SiO_2_ layer. The residual stress in the SiO_2_ layer and the device layer exhibits a gradient along the thickness direction. On the other hand, since the thickness of the underlying silicon is considerably larger than the thickness of the other two layers, the residual stress in the underlying silicon can be approximated as uniformly distributed.

## 3. Mechanical Theoretical Modeling

In the processing of MEMS devices using the SOI wafer structure, the gradual release of the underlying silicon and SiO_2_ layers disrupts the original stress balance mechanism within the SOI structure, leading to the development of residual stress. This residual stress induces bending deformation in the MEMS devices, subsequently impacting their performance and reliability. When processing a cantilever beam using the SOI wafer, initially, before the release of the substrate layer and the SiO_2_ layer in the three-layer SOI structure, the cantilever beam is in an ideal state of stress balance due to the mutual constraint exerted by the layers of the SOI wafer, as depicted in Figure 5a. Since the residual stress σ in the cantilever beam exhibits a stress gradient along the thickness direction, this stress gradient can be approximated as the superposition of stress σ_0_ and σ_1_, as illustrated in Figure 5b.

Prior to the release of the substrate, a shear stress *F*_1_ exists within the SiO_2_ layer to counterbalance the residual stress of the substrate, as depicted in Figure 6a. Since the shear force *F*_1_ is an eccentric force, it generates a stress gradient within the SiO_2_ layer while laterally restraining the layer from stretching. Consequently, after the substrate is released, the influence of the shear force *F*_1_ on the SiO_2_ layer is also eliminated, and the lateral constraint on the cantilever beam at the lower end of the SiO_2_ layer disappears. This results in lateral elongation ${\;\Delta l}_{{SiO}_{2}}$ of the SiO_2_ layer and, concurrently, because of the stress gradient, a moment is generated, leading to deformation *y*_1_ as illustrated in Figure 6b.

In the case of a cantilever beam with a SiO_2_ layer, the cantilever beam experiences a shear force *F*_0_ to counterbalance the stress caused by the SiO_2_ layer, as depicted in Figure 7a. As the shear force *F*_0_ is an eccentric force, it induces a stress gradient within the cantilever beam, restraining the beam’s lateral contraction. Consequently, when the SiO_2_ layer is released, the influence of the shear force *F*_0_ on the cantilever beam is also eliminated, removing the lateral deformation constraint within the beam. As a result, the beam undergoes lateral deformation with shortening Δ*l_Si_*, while the stress gradient generates a moment, leading to longitudinal bending deformation *y_2_*, as illustrated in Figure 7b.

According to the above analysis, considering the compressive stress provided by the SiO_2_ layer in the SOI wafer needs to be shared by the substrate layer and the device layer. That is to say, the value of the residual stress in the SiO_2_ layer has a residual stress value that is the sum of the substrate layer and the device layer, so the amount of deformation that occurs when removing the SiO_2_ layer will be greater than the amount of deformation that occurs when removing the substrate layer. Therefore, in the actual process of the MEMS beam, due to the residual stress existing in the SOI wafer, after releasing the substrate and the SiO_2_ layer, the cantilever beam not only shortens laterally but also deflects downward. By establishing the mechanical relationship between residual stress and residual strain of MEMS beams, residual stress in SOI wafers can be better mapped. The schematic diagram of SOI cantilever beam deformation after two different processes is shown in Figure 8.

First, to examine the impact of shear force *F*_0_ on the SiO_2_ layer of the cantilever beam, a SiO_2_ sacrificial layer is incorporated into the downward curved cantilever beam. This addition serves two purposes: it not only induces lateral elongation Δ*l_Si_* but also generates deflection deformation in the cantilever beam, as illustrated in Figure 9. Specifically, the addition of the SiO_2_ layer can be regarded as applying a bending moment *M*_0_ to the cantilever beam, thereby altering the deflection of the beam from downward to upward. Consequently, the resulting deflection is manifested as *y*_1_ and *y*_2_.

The relationship between equivalent moment *M*_0_ and the shear force *F*_0_ can be expressed as
(1)$$M_{0}=F_{0}\times \frac{h}{2}$$
where *h* represents the thickness of the cantilever beam. The deflection *y* and angle *θ* of the cantilever beam caused by the bending moment *M*_0_ are, respectively, expressed as
(2)$$y=\frac{M_{0}l^{2}}{2EI}$$
(3)$$\theta =\frac{M_{0}l}{\;2EI}$$
where *l* represents the length and *I* denote the moment of inertia of the cantilever beam. *E* corresponds to the elastic modulus of the material used for the cantilever beam. When subjected to the shear force *F*_0_, the cantilever beam experiences a stretching effect, resulting in an elongation Δ*l_Si_* in its length.
(4)$$\Delta l_{Si\;}=\frac{F_{0}l}{\;EA}$$
where *A* represents the cross-sectional area of the cantilever beam.

The effect of shear force *F*_1_ from the substrate on the cantilever beam with a SiO_2_ layer can be analyzed. When subjected to shear force *F*_1_, the cantilever beam with a SiO_2_ layer experiences two distinct responses. First, it undergoes lateral shrinkage denoted as ${\;\Delta l}_{{SiO}_{2}}$. Additionally, it undergoes deflection deformation, as illustrated in Figure 10. This deflection deformation can be understood as the application of a bending moment *M*_1_ to the beam structure. As a result, the beam’s deflection changes from an upward direction to a horizontal state, resulting in a deflection of −*y*_1_.

The relationship between the equivalent moment *M*_1_ and the shear force *F*_1_ can be expressed as
(5)$$M_{1}=F_{1}\times \frac{h^{\prime }}{2}$$
where *h*′ represents the thickness of the silicon dioxide layer. The deflection *y*_1_ and angle *θ*_1_ of the beam structure caused by the bending moment *M*_1_ are, respectively, expressed as
(6)$$y_{1}=-\frac{M_{1}l^{2}}{2\bar{E}\bar{I}}$$
(7)$$\theta_{1}=-\frac{M_{1}l}{2\bar{E}\bar{I}}$$
where $\bar{E}\bar{I}$ is the equivalent bending stiffness of the composite beam. The shrinkage length ${\;\Delta l}_{{SiO}_{2}}$ of the silicon dioxide layer due to the shear force *F*_1_ is
(8)$$\Delta l_{SiO_{2}}=\frac{F_{1}l}{E^{\prime }A^{\prime }}$$
where *E*′ represents the elastic modulus of silicon dioxide and *A*′ represents the cross-sectional area of silicon dioxide layer.

Clearly, under the constraint of the substrate and silicon dioxide layer, the cantilever beam top silicon layer can be reverted to a horizontal state from downward deflection, which is equivalent to applying a bending moment *M*_2_ to the cantilever beam, and the resulting deflection is *y*_2_, as shown in Figure 11.

The deflection *y*_2_ and angle *θ*_2_ of the cantilever beam caused by the bending moment *M*_2_ are, respectively, expressed as
(9)$$y_{2}=\frac{M_{2}l^{2}}{2EI}$$
(10)$$\theta_{2}=\frac{M_{2}l}{EI}$$

The deformation of the cantilever beam, characterized by downward deflection and lateral shortening, is illustrated in Figure 12. By incorporating the mechanical deformation analysis of the cantilever beam, the lateral contraction length Δ*l_Si_* and the deflection *y*_2_ of the beam structure can be represented as $\Delta l_{1}$ and $d_{1}$, respectively.
(11)$$\Delta l_{1}=-\Delta l_{si}$$
(12)$$d_{1}=-y_{2}$$

The bending deformation exhibited by the cantilever beam is attributed to its internal residual stresses. As a result, it is assumed that no deformation occurs at the free end of the cantilever beam. Instead, the resulting deformation is compensated by the application of a reverse axial force $F_{B}$ and a bending moment $M_{B}$.This arrangement forms an equivalent system for the original static indeterminate beam, as depicted in Figure 13.

In the equivalent system of statically indeterminate beams, the axial tensile length Δ*l*_2_ caused by axial force *F_B_*, and the deflection *f*_2_ caused by bending moment *M_B_* are, respectively, expressed as
(13)$$\Delta l_{2}=\frac{F_{B}l}{EA}$$
(14)$$d_{2}=\frac{M_{B}l^{2}}{2EI}$$

The axial tensile length Δ*l* and the deflection *f* of the cantilever beam are
(15)$$\Delta l=\Delta l_{1}+\Delta l_{2}=-\Delta l_{si}+\frac{F_{B}l}{EA}$$
(16)$$d=d_{1}+d_{2}=-y_{2}+\frac{M_{B}l^{2}}{2EI}$$

The total length of the beam does not change and the deflection and deflection angle at the right end of the beam structure is zero. Therefore, the corresponding deformation coordination conditions are
(17)Δl=0 d=0

By substituting from Equations (15) and (16) into Equation (17), *F_B_* and *M_B_* can be obtained as
(18)$$F_{B}=\frac{EA\Delta l_{si}}{l}$$
(19)$$M_{B}=\frac{2EI\times y_{2}}{l^{2}}$$

From the axial force *F_B_* and bending moment *M_B_*, the residual stress *σ* can be derived as
(20)$$y=y_{1}+y_{2}$$
(21)$$\sigma =\frac{F_{B}}{A}+\frac{M_{B}\times y}{I_{z}}$$
where *y* represents the distance from any point to the neutral axis.

The approach allows for the conversion of the residual stress induced by deformation at the free end of the cantilever beam into an expression that relates to the axial force and bending moment at the right end of the beam. This establishes a mechanical theoretical model for the residual stress in SOI wafers, which is mapped by the residual strain of MEMS beams. By measuring the bending moment of the MEMS beams at various points during the machining process, the beam’s deformation deflection and deflection angle can be determined at each moment. The residual stress value within the cantilever beam is then linked to the equivalent values of axial restraint and bending moment. Through the measurement of axial restraint and bending moment, it becomes possible to obtain the residual stress present within the SOI chip.

## 4. Experimental Test and Result Analysis

### 4.1. Sample Preparation and Experimental Test

In the experiment, two SOI wafers with both wafer orientations <100> are utilized to fabricate cantilever beam samples. Each SOI wafer consists of a 6 μm top silicon device layer, a 1 μm SiO_2_ layer, and a 380 μm silicon substrate. The Young’s modulus of the SiO_2_ layer as a thin-film material decreases with increasing thickness, and its corresponding Young’s modulus is 88 GPa, 81 GPa, and 77 GPa for SiO_2_ layer thicknesses of 0.5 μm, 1 μm, and 2 μm, respectively [30]. In this work, the thickness of the SiO_2_ layer is 1 μm and Young’s modulus is defined as 81 GPa.

The preparation process for the cantilever beam samples is outlined in Figure 14. For SOI wafer 1, the substrate and SiO_2_ layer of the cantilever beam are removed. This results in a cantilever beam structure with only the top silicon device layer remaining. For SOI wafer 2, the substrate is removed while the SiO_2_ layer is retained. The cantilever beam structure of this wafer consists of the top silicon device layer and the SiO_2_ layer, with the substrate removed.

In the experiment, a confocal laser scanning microscope (CLSM), as depicted in Figure 15, is employed for a deformation scanning of the samples. The scanning device has a horizontal resolution of 0.12 μm and a *Z*-axis resolution of 0.01 μm. Figure 16 illustrates the images of two types of cantilever beams: one with the removal of the SiO_2_ layer and the other without such removal. Both types of cantilever beams exhibit significant residual deformation at the tail end. This residual deformation is characterized by out-of-plane deflection, which leads to blurriness or incompleteness when observed under an optical microscope. To address this, the residual stress of the wafer is characterized and analyzed based on the strain measured from the cantilever beam. The specific process for characterization is outlined in Figure 17.

### 4.2. The Result Analysis

In the experiment, two sets of cantilever beams with the same position coordinates are tested on two SOI wafers. In the first group, there are two types of cantilever beams. The first type, referred to as CNT-1, does not have the SiO_2_ layer and has a length of 2000 μm and a width of 18 μm. The second type, denoted as CNT-2, has the SiO_2_ layer intact and has a length of 2000 μm and a width of 18 μm. In the second group, there are also two types of cantilever beams. The first type, labeled as CNT-3, does not have the SiO_2_ layer and has a length of 2000 μm and a width of 14 μm. The second type, designated as CNT4, has the SiO_2_ layer and has a length of 2000 μm and a width of 14 μm. However, it is worth noting that CNT-4 has experienced fracture at the midpoint, resulting in a remaining beam length of approximately 1000 μm. The structural morphology of the first group of cantilever beams (CNT-1 and CNT-2) has been obtained through CLSM, as depicted in Figure 18.

The residual strain data of the cantilever beams were obtained through experimental measurements, and curve fitting was performed using MATLAB. The deformation curves for the first and second groups of cantilever beams are depicted in Figure 19 and Figure 20, respectively.

Considering that the length of the cantilever beam is 2000 μm, any data beyond this range are considered invalid in Figure 19. In Figure 19a, the deformation curve of the CNT-1 cantilever beam is presented. When the substrate and the SiO_2_ layers are removed, the cantilever beam exhibits a downward deflection deformation. This deflection direction aligns with the predictions of the mechanical theoretical model discussed earlier, with a deflection magnitude of 28.8 μm. The roughness present in the curve mainly stems from the surface roughness of the beam after the SiO_2_ layer removal. Figure 19b illustrates the deformation curve of the CNT-2 cantilever beam. It can be observed that the cantilever beam, with the SiO_2_ layer intact, undergoes an upward deflection deformation. This deflection direction also aligns with the predictions of the mechanical theoretical model, resulting in a deflection of 307.7 μm.

In Figure 20a, the deformation curve of the CNT-3 cantilever beam is presented, exhibiting a downward deflection of 15.14 μm. In Figure 20b, the deformation curve of the CNT-4 cantilever beam is displayed. However, due to the cantilever beam fracture occurring at 1000 μm, data beyond this point are considered invalid. Within the valid data range, the cantilever beam, with the SiO_2_ layer intact, undergoes an upward deflection deformation with a deflection of 80.8 μm.

For the first group of cantilever beams, CNT-1 and CNT-2, the residual stress of the cantilever beam is calculated by combining the experimentally measured residual strain data with the mechanical theoretical model. The relevant mechanical results are summarized in Table 1. Similarly, the residual stress of the second group of cantilever beams is calculated using the same method, and the mechanical parameters’ results are presented in Table 2.

Based on the calculation results mentioned above, the residual stress in the SOI wafer, determined by mapping the residual strain of the two sets of MEMS beams, is found to be 33.03 MPa and 31.45 MPa, respectively. Sasaki et al. [21] established a mechanical model to map residual stresses by utilizing the rotational deformation of a ring frame. They measured the rotational deformation of the ring support frame using a white light interferometer to characterize the residual stresses in SOI wafers, yielding values of approximately 47 MPa. Zhang et al. [22] employed a micro-pointer indication structure to characterize residual stress in a fixed-supported beam structure. The residual stress in the measured structure was deduced from the deflection of the rotating beam caused by the residual stress in the fixed-supported beam structure. They characterized the residual stress in the SOI wafer to be approximately 25 MPa. The values of residual stresses calculated in this paper are in the same order of magnitude and close to those reported in the literature, indicating the feasibility of the proposed method. In addition, compared with other research results, the method proposed in this paper has higher numerical accuracy, and the absolute value of the two sets of measured values is 1.58 MPa, which makes the numerical reference more accurate.

The agreement between the calculated residual stress and the measured residual strain of the cantilever beam in this paper further validates the accuracy and feasibility of the proposed residual stress characterization method for SOI wafers.

The method proposed in this paper for measuring SOI residual stress primarily relies on stable and effective mechanical theoretical models and formulas. Moreover, since the method only requires measuring the residual strain of the cantilever beam, the testing process is simplified, resulting in higher efficiency, lower cost, and easier operation.

In industrial production, the characterization of residual stress in SOI wafers is often challenging. However, the theory proposed in this paper provides a more precise range for determining the value of residual stress, thus offering valuable insights for industrial manufacturing.

## 5. Conclusions

Due to the inconsistent material properties of silicon and silicon dioxide, thermal mismatch stress and intrinsic stress occur when a thin layer of silicon dioxide grows on the silicon surface. The presence of residual stress affects the performance of MEMS devices fabricated using SOI silicon wafers. In this paper, we propose a high-precision, easy-to-implement, and cost-effective characterization method for measuring residual stress in SOI wafers. We analyze the generation mechanism of residual stress in SOI wafers and develop a mechanical theoretical model for MEMS beams. The feasibility of our method is verified through experimental testing and characterization using laser scanning confocal microscopy. This research provides theoretical references and experimental support for the design and optimization of devices based on SOI-MEMS technology.

## Figures and Tables

**Figure 1 micromachines-14-01510-f001:**
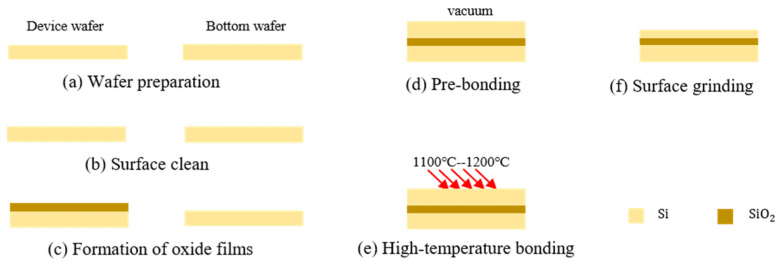
The process of SDB technology for preparing SOI wafers.

**Figure 2 micromachines-14-01510-f002:**
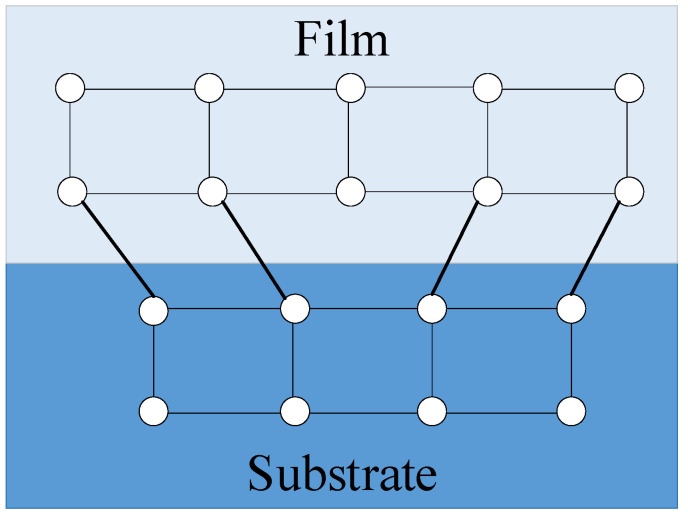
Edge mismatch dislocation due to lattice mismatch.

**Figure 3 micromachines-14-01510-f003:**
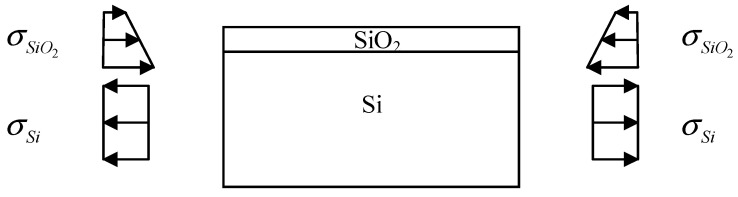
The residual stress distribution of the layer and substrate (edges).

**Figure 4 micromachines-14-01510-f004:**
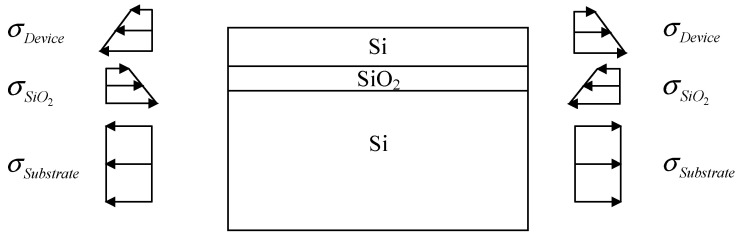
Distribution of residual stress in an SOI wafer (edges).

**Figure 5 micromachines-14-01510-f005:**
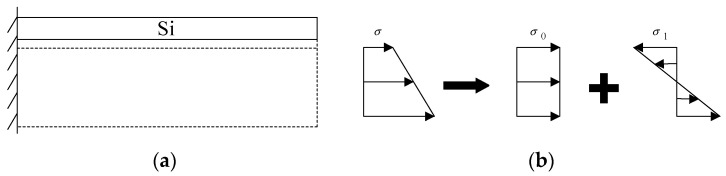
(**a**) The ideal state of a cantilever beam. (**b**) The schematic diagram of gradient stress.

**Figure 6 micromachines-14-01510-f006:**
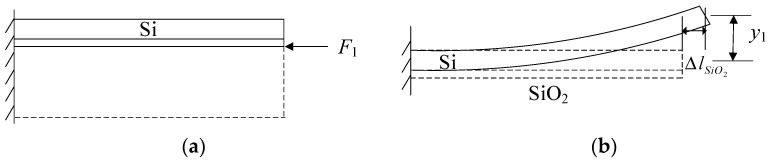
(**a**) The shear force of substrate on the silicon layer. (**b**) The deformation of a cantilever beam with a SiO_2_ layer after releasing substrate.

**Figure 7 micromachines-14-01510-f007:**
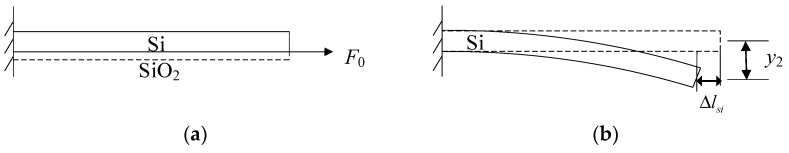
(**a**) The shear force of the SiO_2_ layer on the cantilever beam. (**b**) The deformation of the cantilever beam after releasing the SiO_2_ layer.

**Figure 8 micromachines-14-01510-f008:**
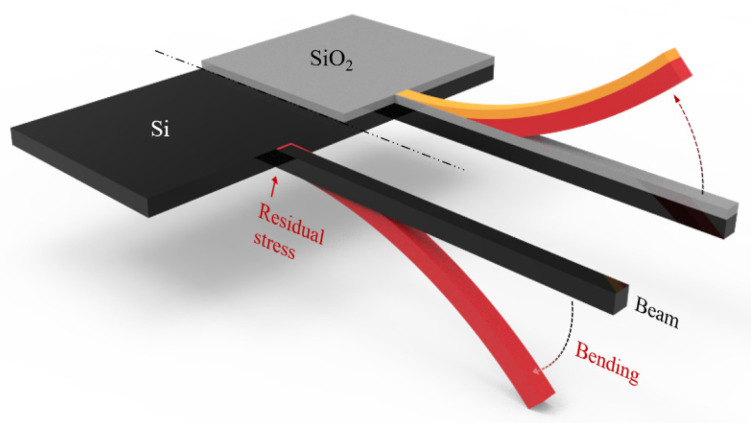
SOI cantilever beam deformation diagram.

**Figure 9 micromachines-14-01510-f009:**
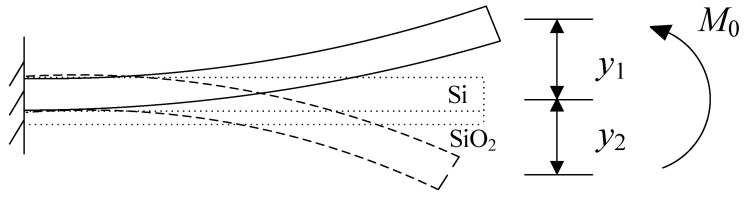
The effect of a silicon dioxide layer on the cantilever beam.

**Figure 10 micromachines-14-01510-f010:**
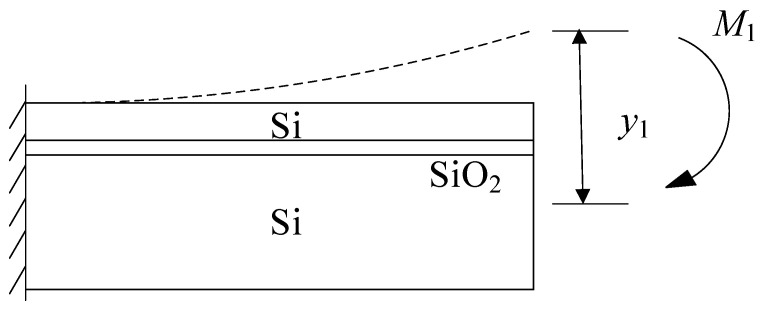
The effect of substrate on the composite beam.

**Figure 11 micromachines-14-01510-f011:**
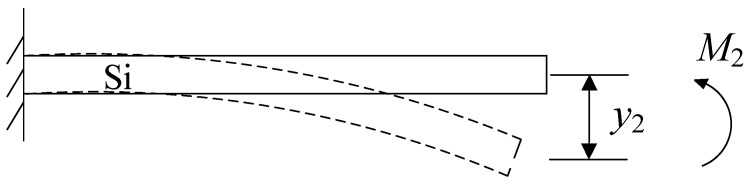
The effect of substrate and silicon dioxide on the cantilever beam.

**Figure 12 micromachines-14-01510-f012:**
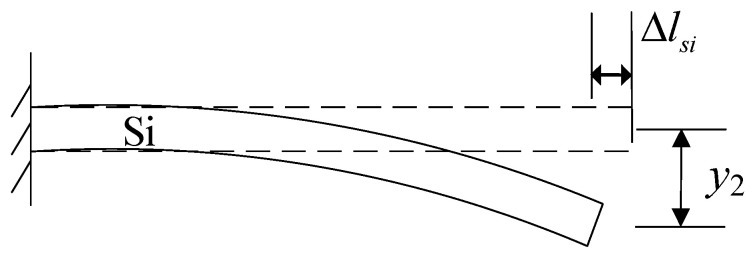
Deformation analysis of the cantilever beam.

**Figure 13 micromachines-14-01510-f013:**
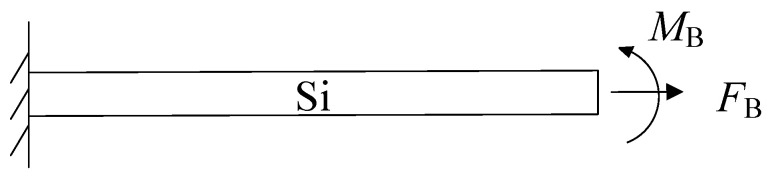
The equivalent system of the statically indeterminate beam.

**Figure 14 micromachines-14-01510-f014:**
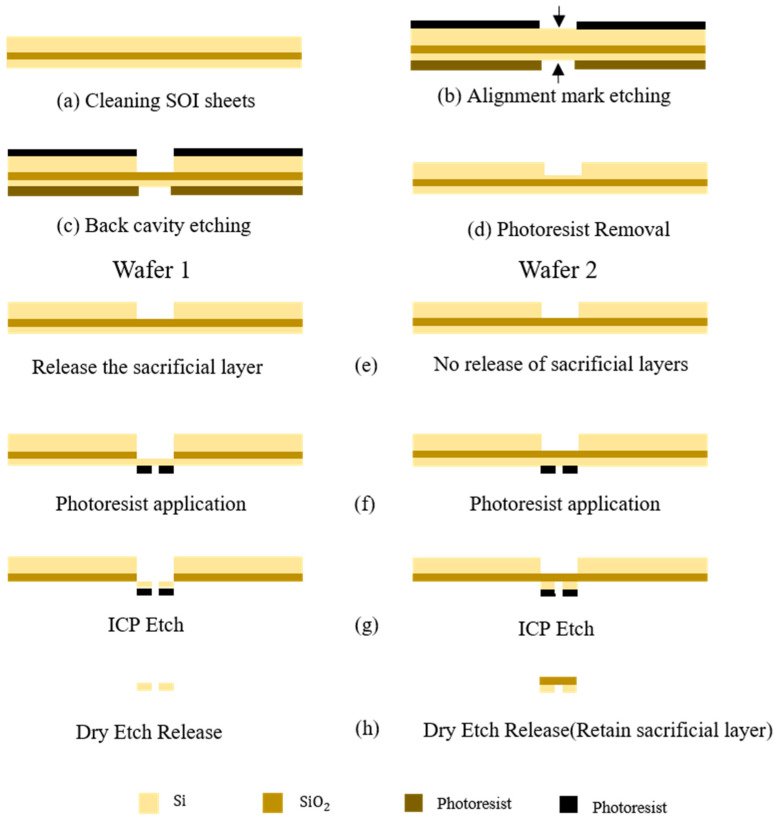
Schematic diagram of the preparation process for two SOI wafers.

**Figure 15 micromachines-14-01510-f015:**
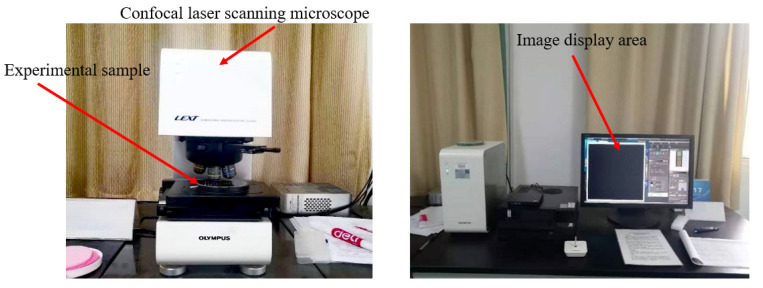
Physical image of confocal laser scanning microscope.

**Figure 16 micromachines-14-01510-f016:**
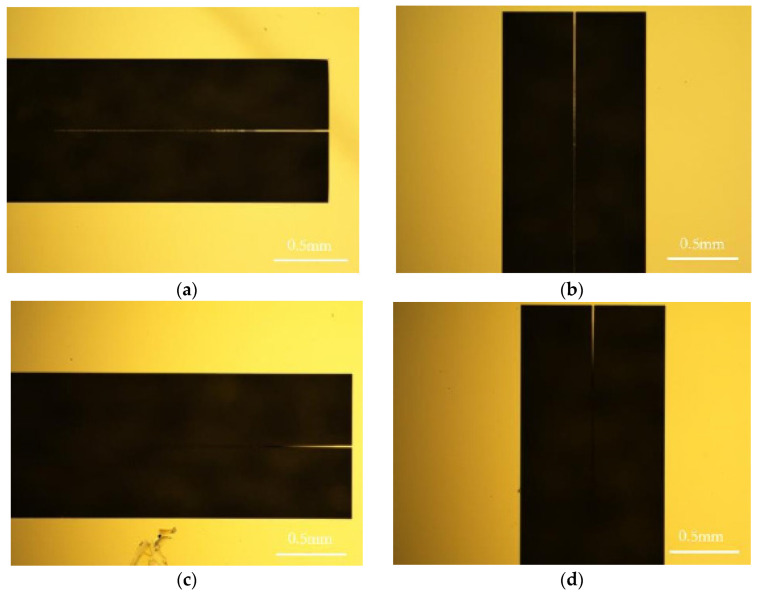
Images of two types of cantilever beams: (**a**) transverse cantilever beam (no sacrifice layer released), (**b**) longitudinal cantilever beam (no sacrifice layer released), (**c**) transverse cantilever beam (sacrificial layer released), and (**d**) longitudinal cantilever beam (sacrificial layer released).

**Figure 17 micromachines-14-01510-f017:**
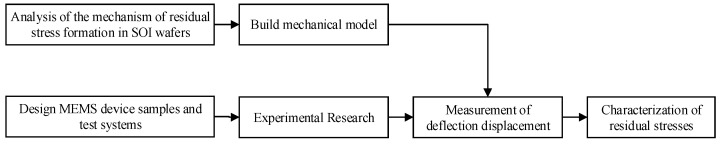
Schematic diagram of residual stress characterization process.

**Figure 18 micromachines-14-01510-f018:**
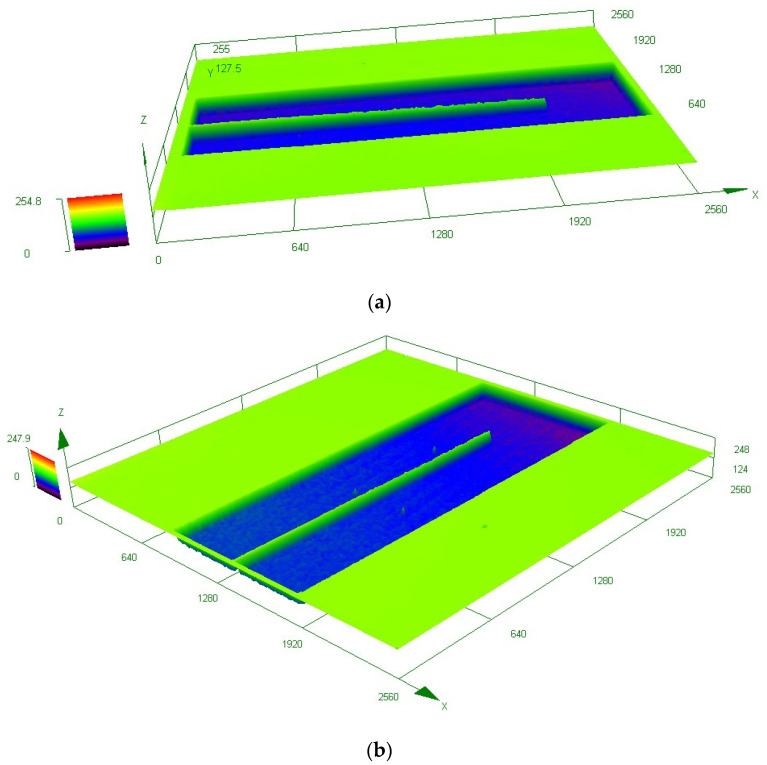
The 3D morphology of the first group of cantilever beams: (**a**) CNT-1 and (**b**) CNT-2.

**Figure 19 micromachines-14-01510-f019:**
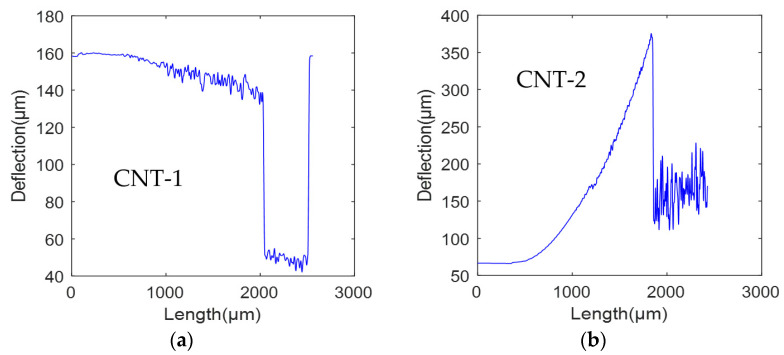
Beam deformation curves of the first group of cantilever beams: (**a**) CNT-1 and (**b**) CNT-2.

**Figure 20 micromachines-14-01510-f020:**
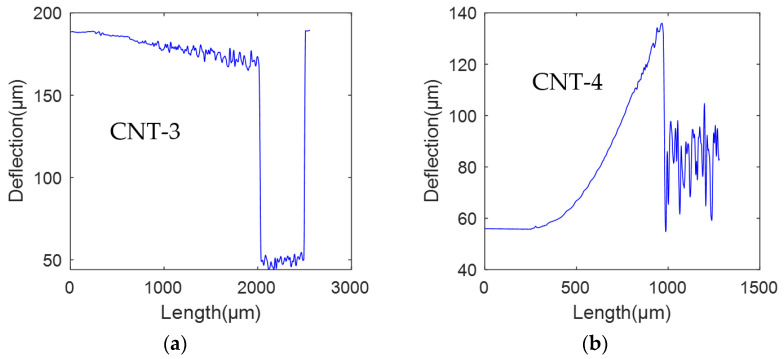
Beam deformation curves of the second group of cantilever beams: (**a**) CNT-3 and (**b**) CNT-4.

**Table 1 micromachines-14-01510-t001:** The mechanical parameters’ results of the first group cantilever beams.

Mechanical Parameters	Symbol	Value
Deflection of composite cantilever beam	*y* _1_	307.7 μm
Deflection of cantilever beam	*y* _2_	28.8 μm
Shortened length of cantilever beam	Δ*l_Si_*	0.337 μm
Restrained axial force of clamped beam	*F* _B_	28.39 N
Restrained bending moment of clamped beam	*M* _B_	7.28 × 10^−10^ N·m
Residual stress	*σ*	33.03 MPa

**Table 2 micromachines-14-01510-t002:** The mechanical parameters’ results of the second group cantilever beams.

Mechanical Parameters	Symbol	Value
Deflection of composite cantilever beam	*y* _1_	80.08 μm
Deflection of cantilever beam	*y* _2_	10.1 μm
Shortened length of cantilever beam	Δ*l_Si_*	0.141 μm
Restrained axial force of clamped beam	*F_B_*	18.48 N
Restrained bending moment of clamped beam	*M_B_*	7.94 × 10^−10^ N·m
Residual stress	*σ*	31.45 MPa

## Data Availability

No new data were created for this work.

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
