# Peer review of "Characterization of Residual Stress in SOI Wafers by Using MEMS Cantilever Beams"

_micromachines, 2023, doi:10.3390/mi14081510_

Round 1

Reviewer 1 Report

This paper addresses an excellent topic that is rarely addressed in MEMS analysis and manufacturing. I recommend that the paper be published, after making the following minor revisions:

. On page 3, line 113, change “since” to “because”.

. Fix the inconsistent font of the text in the caption of Figure 7.

. On page 6, line 185, change “Firstly” to “First”.

. On page 6, line 210, change “Firstly” to “First”.

. On page 9, line 268, change “By substituting equation (15) and (16) into equations (17)” to “By substituting from equation (15) and (16) into equation (17)”

. There is a problem in Figure 15. The arrows are pointing to blank areas. This figure should be revised.

Only some minor editing of English is required as mentioned above.

Author Response

Thank you for your suggestion, the response to the comments are in the attachment. Please see the attachment.

Reviewer 2 Report

1- The abstract needs to be modified to highlight the originality of the manuscript instead of providing generic details.

2- Kindly describe the challenges of previous non-destructive measurement methods more clearly (line 67).

3- Kindly cite a reference for Figure 1.&2. if it is not your own work.

4- Kindly describe the reason (line 175).

5- Please ensure that the format remains consistent (line 235).

6- Please take into account the impact of size on your analysis. The Young's modulus of a thin film material varies depending on its size.

7- Please provide a quantitative assessment of your comparison with references 21 and 22. Additionally, describe the benefits of the proposed method over these references in measurable terms, such as accuracy.

8- Please take into account the formatting of Figure 19 and 20.

There are a few grammatical errors, for example, in line 369, "is" should be changed to "are" in the phrase "When the substrate and the SiO2 layer (layers) is (are) removed...". It is recommended to thoroughly review the article and rectify any English language errors.

Author Response

(The authors gave the same response as above.)

Round 2

Reviewer 2 Report

The quality of Figure. 18 must be improved and units must be included. 

Figure 20 has formatting issues. 

There are some grammatical mistakes, for instance in line 161. The sentence "According to the above analysis, considering that the...." contains an unnecessary word "that" which should be eliminated.

Author Response

Thank you for your suggestions. We have corrected some issues in the text and provided responses to the review report. Please see the attachment.
